# Addressing Food Insecurity during the COVID-19 Pandemic: Intervention Outcomes and Lessons Learned from a Collaborative Food Delivery Response in South Florida’s Underserved Households

**DOI:** 10.3390/ijerph19138130

**Published:** 2022-07-02

**Authors:** Nana Aisha Garba, Lea Sacca, Rachel D. Clarke, Prasad Bhoite, John Buschman, Virama Oller, Nancy Napolitano, Samuel Hyppolite, Sophia Lacroix, Al Archibald, Ocean Hamilton, Tobi Ash, David R. Brown

**Affiliations:** 1Department of Humanities, Health and Society, Herbert Wertheim College of Medicine, Florida International University, Miami, FL 32413, USA; lsacca@fiu.edu (L.S.); rclarke@fiu.edu (R.D.C.); pbhoite@fiu.edu (P.B.); violle@fiu.edu (V.O.); nnapolit@fiu.edu (N.N.); shyppoli@fiu.edu (S.H.); slacroix@fiu.edu (S.L.); drbrown@fiu.edu (D.R.B.); 2Chaplin School of Hospitality and Tourism Management, Florida International University, Miami, FL 33181, USA; jbuschma@fiu.edu; 3Grace United Community Church, Miami, FL 32413, USA; aarchibald@graceunitedlearning.com; 4Redland Ahead Inc., Farmer to Families Program, Homestead, FL 32413, USA; ocean.rcfam@gmail.com; 5Joshua’s Heart Food Pantry, North Miami Beach, FL 32413, USA; tobi.t.ash@gmail.com

**Keywords:** COVID-19, pandemic, social determinants of health, food insecurity, community outreach, community engagement, underserved community

## Abstract

Background: The COVID-19 pandemic highlighted underlying disparities in health, healthcare access, and other social factors that have been documented for racial/ethnic minorities. The social-distancing mandate exacerbated the impact of social determinants of health, such as unemployment and food insecurity, particularly among underserved minority populations. We highlight intervention outcomes and lessons learned from the Florida International University (FIU) Herbert Wertheim College of Medicine (HWCOM) NeighborhoodHELP’s response to pandemic-related food insecurity among Miami Dade County’s underserved population. Methods: Following the stay-at-home mandate, a weekly needs assessment of program households was conducted by the NeighborhoodHELP team, during which food insecurity emerged as a pandemic-related urgent need, rising from three percent of program Households in March 2020 to 36.9 percent six months later. Consequently, the program staff collaborated with another FIU department, community partners, and a benefactor to develop a food donation and delivery project. Results: Fifteen hundred and forty-three culturally appropriate food boxes were delivered to 289 participating households, comprising 898 household members, over a 14-month period. Conclusion: This project underscores the importance of leveraging community assets to address their needs during a crisis and the significance of sustained community engagement for researchers and service providers who work in underserved communities.

## 1. Introduction

The coronavirus disease 2019 (COVID-19) pandemic is unparalleled in modern times, with a high rate of infection and fatality [1]. It has brought to light underlying disparities in health, healthcare access, living conditions, financial security, and other social factors that have long been documented for racial/ethnic minorities [2]. While the physical impacts of the COVID-19 infection are severe, the socioeconomic impacts are among the most profound since the Great Depression [3]. As a result of social distancing and the stay-at-home mandates in place during the first wave of the pandemic, a significant number of people were either furloughed or lost their jobs and could no longer afford the basic necessities of life [4]. Additionally, there was a shutdown of public transportation, making movement almost impossible for people who relied on the bus or other forms of public transportation to carry out their daily activities, such as working, grocery shopping, and visiting friends and family [4]. People were generally living in fear of the virus and would not venture out even if they could; socialization was down to the minimum, so along with fear, there was an increase in the number of people with depression and anxiety, loss of social support, and social isolation [5].

Food insecurity, a social determinant of health (SDOH) linked to malnutrition and the lack of sufficient access to food for an active and healthy life due to physical, social, and economic barriers, was significantly exacerbated across the country and globally [6,7]. Food production, distribution, and storage processes were negatively impacted by control measures created to limit the spread of COVID-19 [8,9,10]. The early stages of the pandemic witnessed a reduction in food availability due to supply chain interruptions, decreased agricultural practices, and the stockpiling of food items by individuals who could afford to do so [8,9,11]. Additionally, due to the truncation of global transportation, the demand for food outweighed the supply, resulting in an increase in food prices, making food unaffordable for many and increasing the number of people who could not meet the recommended daily nutritional intake [12]. Existing safety nets, such as food banks and Supplemental Nutrition Assistance Programs (SNAPs), were failing to keep up with the increase in emerging needs [13,14,15,16].

Furthermore, the implementation of COVID-19-related government measures on mobility restriction resulted in an increased loss of employment in both formal and informal sectors, which decreased the capacity of vulnerable populations to access food from their regular channels, including daily markets, restaurants, street vendors, and school-based food programs [17,18,19,20]. The ability to purchase healthy and nutritious food was impacted by the spike in prices and the need to adapt and feed an entire household during an ongoing pandemic [10,21]. Low-income neighborhoods and families replaced fresh and nutritious fruits and vegetables with shelf-stable foods and emergency food rations as coping mechanisms in response to the worsening food insecurity crisis inflicted by the pandemic [21,22,23]. Children from low-socioeconomic-status homes who depended on school-provided free meals for daily nutrition were severely affected when schooling switched to a remote platform as it limited their ability to access enough food to eat [24]. Older populations were especially affected with the closing of senior centers and modifications to delivery programs such as Meals on Wheels [24,25]. The pandemic therefore heightened the level of food insecurity among vulnerable, low-income households, compounding challenges associated with poverty, unemployment, and limited transportation, with the potential to lead to long-term malnutrition along with its associated negative health outcomes [10,20].

In 2018, prior to the pandemic, 11% of households in the United States experienced food insecurity [26]. However, in the initial months of the pandemic, studies show the rate more than tripled with 35–38% of US households reporting food insecurity [27,28]. This is particularly alarming because of the well-documented untoward health impact of food insecurity. For example, during the pandemic, studies reported an association between food insecurity and poorer mental health outcomes, such as depression and anxiety [29,30]. In fact, according to a study by Fang et al., during the pandemic, food insecurity was associated with a 257% higher risk of anxiety and 253% higher risk of depression, which was eight times the risk of developing anxiety and more than nine times the risk of developing depression following a job loss during the pandemic [29]. Additionally, adults who are food insecure have been shown to have higher rates of obesity; chronic diseases such as diabetes, hypertension, and stroke; and health disparities [31,32], while children who are food insecure face a higher risk of cognitive behavioral, developmental and mental health problems compared to their food-secure peers [33,34,35].

For the past decade, the Florida International University (FIU) Herbert Wertheim College of Medicine (HWCOM), through its flagship Green Family Foundation Neighborhood Health Education Learning Program (NeighborhoodHELP, Miami, FL, USA), has implemented an innovative service-learning, community outreach program, geared towards addressing social determinants that affect the health outcomes of household members in Miami Dade County’s underserved communities. The program has three arms, namely curricular, clinical, and outreach. The outreach arm comprises a team of outreach workers (community health workers) whose role is to engage community-based organizations and other stakeholders with the primary aim of encouraging them to refer medically and socially underserved households in their care to the program. Post referral, each household is assigned to an outreach worker who vets and enrolls them into the program, if eligible, and then navigates them to community-based resources in response to their identified needs. It is the duty of each outreach worker to establish, nurture, and maintain relationships with members of their assigned households on behalf of the program.

The other two arms of the program are the clinical and the curricular. Through the clinical arm, uninsured members of NeighborhoodHELP receive free, preventive healthcare and behavioral health services at one of the program’s mobile health centers often situated weekly within targeted vulnerable communities. The curricular arm of the program is made up of inter-professional student teams assigned to enrolled households (one per team). They are tasked with conducting multiple annual team visits under clinical faculty supervision for the purpose of providing longitudinal household-centered care focused on identifying and mitigating social determinants that can impact the health outcomes of members of the households. Prior to the pandemic, each household enrolled in the program received a minimum of one monthly check-in phone call from either their assigned outreach worker or a medical student from their assigned student team.

The following study will highlight the outcomes of the NeighborhoodHELP’s community needs assessments, as well as the implementation and lessons learned from the food delivery intervention developed in response to the food insecurity reported by its members during the pandemic.

## 2. Methods

### 2.1. Early Stages of Food Delivery Response

Upon the declaration of a stay-at-home mandate, knowing how socially and economically vulnerable program participants were, NeighborhoodHELP instituted weekly phone call check-ins and needs assessments of its household members, carried out by the NeighborhoodHELP outreach team, the faculty, and students. During each call to a household, the caller often spoke with the designated head of household, who was usually the more accessible parent or partner. To determine if a household was experiencing food insecurity, the following question was asked: “*Are you experiencing difficulty accessing food as a result of COVID-19?*” Those who answered “yes” were considered to be food insecure. Household members who stated that they were receiving SNAP benefits were asked the following questions to determine if they were food insecure: “*Are your SNAP benefits enough for you, and are you able to buy healthy food?*” Those who answered “no” were considered food insecure. Heads of households were also asked about employment, access to transportation, household income, access to technology, and several other questions designed to assess their COVID-19-related needs. These efforts led to the identification of several pandemic-related urgent needs, chiefly employment, food, and COVID-19 testing and education. In the first 2 months of the pandemic (March and April), there was a 3.2 percent increase in the number of household members who stated that they were either unemployed or underemployed, peaking at a 5.8 percent increase by the fifth month. While the team was able to partner with community-based organizations and stakeholders to develop a comprehensive resource list of hiring organizations and testing locations in their neighborhoods, in addition to providing COVID-19-related health education, the food insecurity issue remained a major barrier to the address. In fact, within the first two months of the pandemic (March and April), the percentage of NeighborhoodHELP household members who reported food insecurity doubled from 3.5 to 8.0 percent. This increase continued and peaked at 36.9 percent between September and October 2020 (Figure 1). In March 2020, NeighborhoodHELP was ill-equipped to address the increased level of food insecurity experienced by some of its participants. Prior to the pandemic, 3.5% of NeighborhoodHELP household members had reported food insecurity and were given a list of food pantries close to their homes where they could gain access to food. However, during the pandemic, only people with cars could access food pantries, which had transitioned to a “drive-through” model to reduce the risk of spreading the virus. Since a significant proportion (45.2%) of members enrolled in the program had transportation challenges, including the lack of personal cars, they could no longer be referred to food pantries. In addition, people who had cars would start queuing at dawn and leave hours later when the food ran out, often without any food and with much less gas in their cars than they started off with.

### 2.2. Funding

To address this increasing need, the program leadership convened a series of meetings with a benefactor, several NeighborhoodHELP community partners, the outreach team, and representatives of the other two arms of the program to brainstorm ways to address this issue. Consequently, the program received a financial donation from one of its benefactors, and it was decided that NeighborhoodHELP would purchase food that would be delivered by members of its outreach team who were no longer conducting in-person household visits due to the social distancing mandate.

### 2.3. Building Partnerships and Involving Stakeholders

Having no prior experience of running a food pantry and distribution site, the outreach team gathered preliminary information about how to create a food pantry by researching best practices online and consulting with an established local food pantry and distribution site (Joshua’s Heart) that was a NeighborhoodHELP community partner. One of the program’s community partners, a faith-based organization (Grace United Community Church), provided a fully air-conditioned location which served as a food storage, boxing, and dispatch location within the community (Figure 2). Food was purchased weekly from local grocery stores and transported to the distribution location in bulk by the NeighborhoodHELP outreach team (Figure 3). The program also partnered with the faculty and staff from FIU’s Chaplin School of Hospitality, who provided guidance on how to estimate the amount of food to be delivered to each household based on the number of household members, and how to properly store food to avoid waste and ensure that it reached the consumer in an optimum condition. Another collaboration was with a United States Department of Agriculture-sponsored farm-share project (Redland Ahead inc. Farmer to Families Program) to obtain fresh fruits and vegetables, which were also distributed biweekly to households. Owing to the fact that NeighborhoodHELP outreach workers were hired from the communities that the program served, they were pivotal in helping to select culturally appropriate foods that were distributed to our predominantly Hispanic, Haitian, and African American households. Since rice and beans are the two main staples in Haitian and Spanish households, we ensured that food boxes going to these households contained dry rice and beans, some Spanish or Haitian condiments and spices, and vegetable oil since most households would prefer to cook their own food. Each box was also accompanied by a complementary box of fruits and vegetables based on availability.

## 3. Results

At the onset of the pandemic in March 2020, 860 underserved households (2342 household members) were actively enrolled in NeighborhoodHELP with an average household size of 2.73 including children. Of the 860 households enrolled in the program, 219 of them have 389 children, making the average number of children per household 1.79.

Descriptive statistics were used to categorize households in need based on their sociodemographic characteristics (age, gender, race/ethnicity, and annual household income) (Table 1). Early in the pandemic, approximately 10 percent of NeighborhoodHELP household members stated that they were receiving some supplemental assistance (SNAP or food bank) to meet their nutritional needs. The first day of the food delivery was on 15 April 2020 (Figure 4). Food boxes were subsequently delivered biweekly (Figure 5) either until the project ended or when the household members indicated that they no longer were in need. Members were encouraged to reach out to their assigned outreach workers if they had special dietary needs or suggestions on how we could improve the quality of the food delivered. While the overall response was one of gratitude, all feedback was seriously considered and promptly addressed where possible. Though students could not deliver food due to the restrictions in place at the time, they contributed to the process by calling households to ensure that somebody was available to pick up the food when it was dropped off and assisted the outreach team with the boxing of food to the specification of each household in preparation for delivery.

The project lasted for 14 months from April 2020 to June 2021, during which NeighborhoodHELP outreach workers purchased, packed and delivered 1543 culturally appropriate food boxes to 289 participating households with 898 household members. Twelve months into the project, the decision was made to not request more funds and to wrap up the COVID-19 food response project based on the following factors: (a) The social distancing mandate had been lifted and businesses were beginning to reopen, resulting in the inevitable loss of our food storage and dispatch space, which had reverted back to its pre-pandemic status, and (b) following the reopening of businesses, there was an increase in the availability of food distribution at other sites, which made our site less necessary, allowing us to revert back to our usual practice of referring people to community resources. However, the project did not wind down abruptly. It was a deliberate process which lasted for about two months because recipients had to be informed ahead of time that they would no longer be receiving food from the program after a specific date, giving them time to process the information, ask questions, and make alternative arrangements. We also used that time to put together a small in-house food pantry for emergencies and to create a list of food pantries by neighborhood which was shared with every household prior to and on the final day of food delivery.

## 4. Discussion

### 4.1. Lessons Learned

Several lessons were learned as NeighborhoodHELP mobilized to meet the increasing food insecurity experienced by its enrolled households during the pandemic. One major lesson was the need for researchers and service providers to remain engaged with the members of the communities they worked with. FIU and HWCOM have, over the years, prioritized sustained community engagement which is one of the reasons why NeighborhoodHELP was created. Prioritizing community engagement in this case means that we are trusted by the community and have access to the infrastructure required to serve them on short notice. This made it easy to assess people’s needs during the crisis and know which stakeholders to invite to the table for guidance on how to best address identified community/household needs, thereby significantly reducing the turnaround time when it was imperative to do so.

Another lesson was that involving community members and stakeholders in all phases (planning, implementation, and conclusion) of a community-based project is essential to ensure the success of the project. In this project, community partners’ involvement meant we did not have to pay for a storage location and the cost of keeping the air conditioning running for a whole year! Moreover, the fact that the space donated by our partner was in the community also significantly cut the potential cost and time of food delivery. Indeed, contributions by our partners enabled us to significantly cut down on overhead costs and spend more on food. Our partnership with the farm-share project also allowed us to stretch the value of the dollar and improve the quality of the food we delivered. The importance of such partnerships has been highlighted in communities with similar demographics, where the most commonly reported implementation strategy to induce change in the community was “building partnerships to support program implementation” [36]. For instance, a similar strategy was successfully employed in an ethnic minority community that was significantly affected by the pandemic, resulting in a depletion of resources. In the American Indian/Alaskan Native (AI/AN) community, based on community input, sexual health experts realized this and had to alter their roles at the early stages of the pandemic to focus on food relief and building water sanitation stations to address the food insecurity crisis spreading across AI/AN households [37]. By ensuring community involvement in the adaptation, dissemination, and evaluation of health programs, along with adopting a comprehensive approach to understanding the emergent needs of the community, food relief efforts were successful in alleviating the challenges experienced by the AI/AN households [37]. We also learned that SDOH do not operate in isolation and should therefore not be addressed without taking other extenuating factors into consideration. In this case, the rising unemployment rate and dwindling access to transportation, in addition to the COVID-19 mitigating measures put in place by the Government, played a key role in contextualizing the growing food security challenges identified in the communities we serve and how they were ultimately addressed. It is important to note that each community is unique in its challenges, existing policies, and the availability of resources that can be leveraged to address the needs of its members [38]. For example, the early pandemic policy landscape in Miami-Dade made food insecurity an urgent concern at the time. In order to address each community’s need, regardless of its challenges, whether it is during a pandemic, a weather-related event, or an economic downturn, a successful intervention should include the following key elements: (1) Prior knowledge of the community and its stakeholders; (2) Familiarity with community assets and infrastructures that can be leveraged to provide assistance to members of the beleaguered community; (3) The capacity to conduct a rapid, ongoing community needs assessment before intervention. In other words, give them what they need, not what you think they need; (4) Flexibility and the knowledge that one does not have all the answers; and (5) The willingness to include members of the community in the planning, implementation, and debriefing process [36,37,38,39,40].

Finally, delivering culturally appropriate food ensured that our families not only received sustenance in their time of need but also the comfort of eating something familiar in a world that had suddenly become unfamiliar and certain. One should approach such situations with some cultural humility.

### 4.2. Limitations

Although the reported rate of food insecurity among NeighborhoodHELP households had dropped from its peak at 36.9% to 23% by the time the project ended (see Figure 1), it is interesting to note that the food insecurity rate had not returned to its pre-pandemic level. Even though the social distancing mandate had been lifted and people appeared to be going about their businesses as usual, the overall quality of life was still a long way from what it was pre-pandemic, especially with the spike in the cost of housing and the increasing cost of living due to inflation. Moreover, not everyone who became unemployed or furloughed at the beginning of the pandemic was able to get their jobs back. This was especially true for racial/ethnic, socioeconomically vulnerable communities, which includes the population that NeighborhoodHELP serves. For these populations, unless underlying factors, such as unemployment, low income, immigration status, and unreliable transportation, which worsened the level of food insecurity during the pandemic, are ameliorated, food insecurity levels will continue to hover above pre-pandemic levels. Another limitation was the tendency of the project team to prioritize shelf-stable foods over fresh foods to reduce cost and minimize waste, thereby sometimes sacrificing quality to stretch the dollar.

## 5. Conclusions

This project underscores the value of leveraging community assets to address needs during a crisis, as well as how sustained community engagement empowers researchers and service providers who work in underserved communities. Future studies should explore the experiences of NeighborhoodHELP participants who experience food insecurity, the main challenges they encountered during the pandemic, and the benefits or challenges of their enrollment in the program to learn from the participants’ perspectives what needs to be improved in the delivery of services to household members and their communities.

## Figures and Tables

**Figure 1 ijerph-19-08130-f001:**
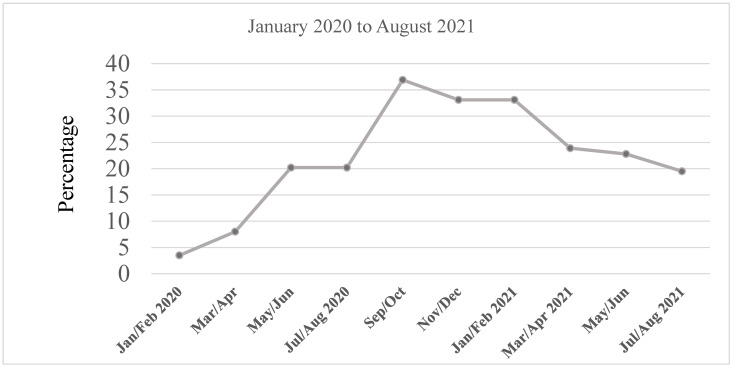
Food Insecurity Trend among NeighborhoodHELP Households.

**Figure 2 ijerph-19-08130-f002:**
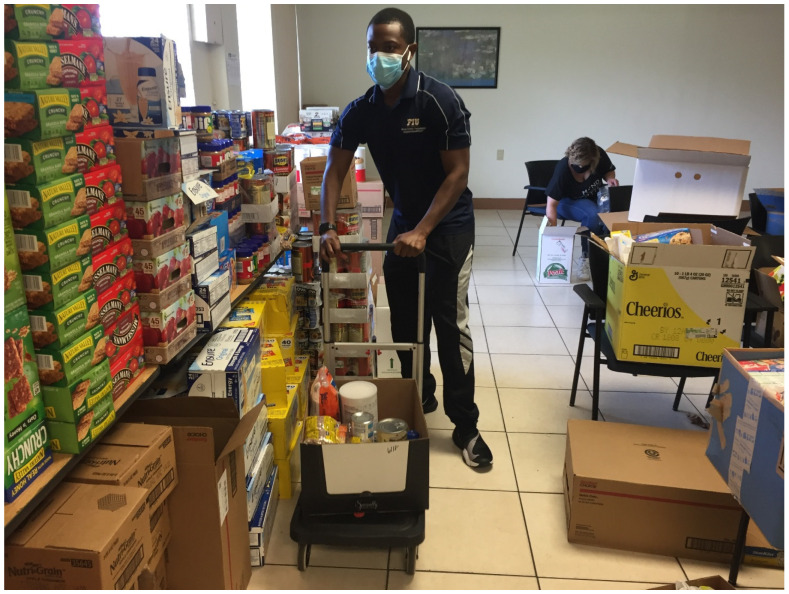
Food Packaging Distribution Location.

**Figure 3 ijerph-19-08130-f003:**
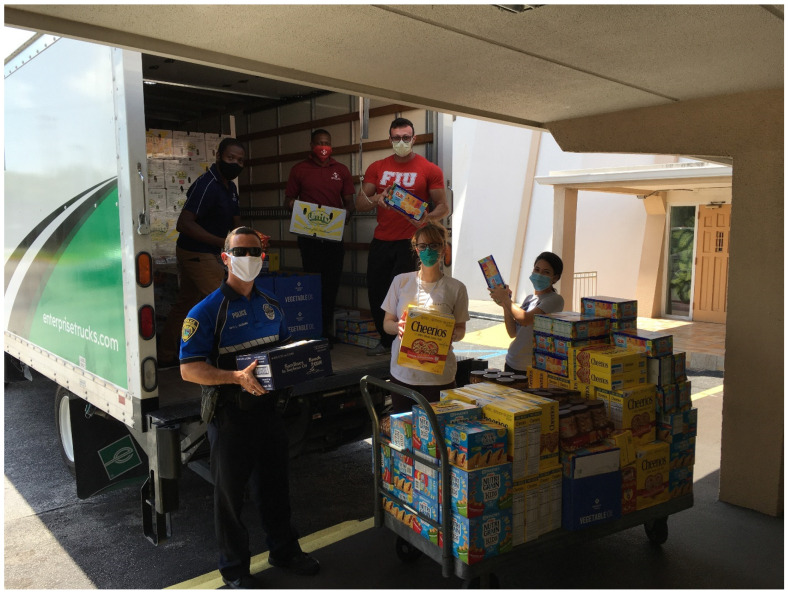
Team Heading to the Food Distribution Site.

**Figure 4 ijerph-19-08130-f004:**
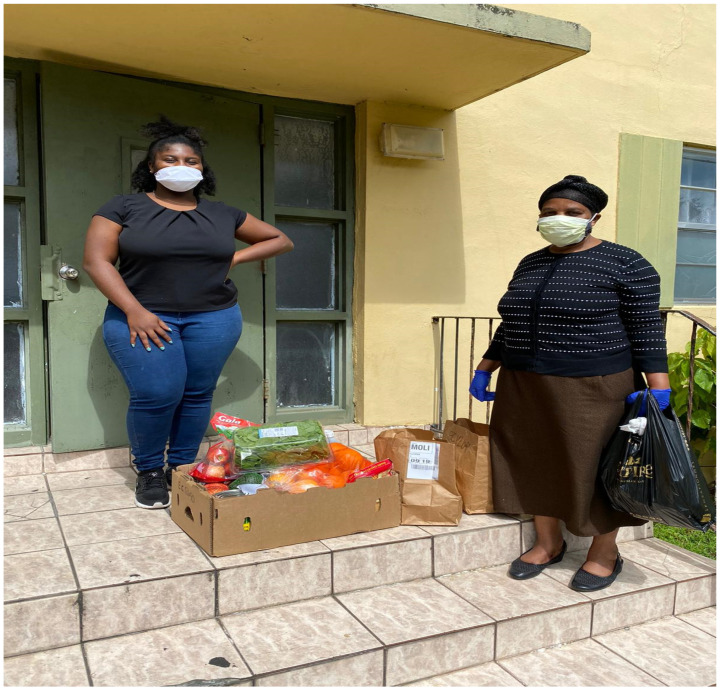
Food Delivery to a NeighborhoodHELP Household.

**Figure 5 ijerph-19-08130-f005:**
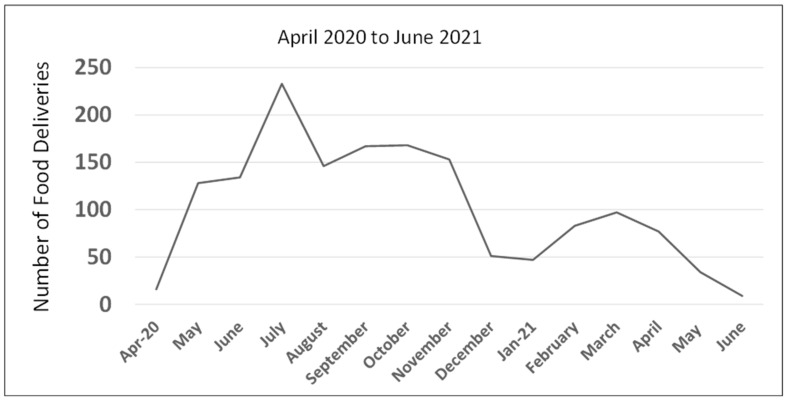
Number of Food Deliveries by the NeighborhoodHELP Team per Month.

**Table 1 ijerph-19-08130-t001:** Demographic Distribution of NeighborhoodHELP Household Members.

Characteristic	N = 2342	
	Count (n)	Percentage (%)
**Age**		
0–17 years	389	16.61
18–34 years	540	23.06
35–44 years	315	13.45
45–54 years	355	15.16
55–64 years	405	17.29
65 and older	338	14.43
**Gender**		
Female	1406	60.00
Male	904	38.25
Undisclosed	6	0.31
**Race**		
White	1075	45.90
Black/African American	969	41.40
Multiracial	159	6.80
Asian	39	1.70
American Indian/Alaska Native	8	0.30
Native Hawaiian & Other Pacific Islander	1	0.00
Unreported/Refused to Report Race	181	7.80
**Ethnicity**		
Hispanic/Latino	1276	54.50
Non-Hispanic/Latino	1015	43.30
Unreported/Refused to Report Ethnicity	51	2.2
**Characteristic**	**N = 860 Households**	
	**Count (n=)**	**Percentage (%)**
**Annual Household Income**		
Under $10,000	600	69.77
$10,000–$19,999	122	14.19
$20,000–$29,999	79	9.19
$30,000–$39,999	35	4.07
$40,000–$49,999	13	1.51
$50,000–$74,999	10	1.16
$100,000–$149,999	1	0.12

## Data Availability

The data presented in this study are available on request from the corresponding authors. The data are not publicly available for participant privacy purposes.

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
