# Peer review of "Addressing Food Insecurity during the COVID-19 Pandemic: Intervention Outcomes and Lessons Learned from a Collaborative Food Delivery Response in South Florida’s Underserved Households"

_ijerph, 2022, doi:10.3390/ijerph19138130_

Round 1

Reviewer 1 Report

The paper is within the scope of the Journal and raises a very interesting issue, related to the much-needed programme activities for underserved households during the Covid-19 pandemic. However, significant editorial changes are needed in the paper as it is not tailored to the requirements of the Journal. I have included detailed comments below.

8-13 (Please complete city, state and country. Indicate correspondence author.)

14-39 Abstract (200 words maximum, without headings)

26 FIU (Please explain the abbreviation.)

40-41 Keywords (in lower case (except COVID-19), separated by semicolons)

43 (Section title should not be left as the last line on the page.)

48 Bailey et al., 2017 and all other references (References should be numbered in the order they appear in the text and in square brackets.)

95 (If a subsection is singled out, there should be at least one more, otherwise no subsection should be singled out.)

124 ., > .

130 Table 1 > Table 1 in Supplementary Material (Please also add the phrase “Supplementary Material” in Supplementary Material. Please also explain, why numbers in Methods in the text and in Material are different, i.e.: 860 households / 2342 household members and 924/1945, respectively.)

126, 153, 166; 217, 245 (Subsections should be numbered, i.e. 2.1, 2.2 and 2.3; 4.1 and 4.2.)

127-141, 229-244, 261-265 (The font is too small.)

142 Figure 1 (Please don’t bold.)

Figure 1 (The word “Percentage” is partially invisible.)

165, 185, 215 (Please cite Figure 2, Figure 3 and Figure 4 in the main text.)

165 (Please don’t bold figure title, i.e. “Food Packaging Distribution Location”.)

266-267 Figure S1: title, Table S1: title, Video S1: title > Table 1: Demographic Distribution of NeighborhoodHELP Household

285 (Please complete this Reference.)

286-360 References (Year should be placed at the end of reference. Journal titles should be abbreviated.)

Reviewer 2 Report

This is a nice description of a very interesting program, illustrated with pictures, however, it has very limited scientific content and its contribution to the literature is not clear.

My comment was only a polite statement for the authors, as the manuscript does not meet the expectations of a scientific publication, as there is no research question, there is no methodology (the Methods part is a simple descripive section) and the results are only the summary of the project implementation. 

The manuscript might fit to a case study collection, but not to a peer reviewed scientific journal.

Author Response

This is a nice description of a very interesting program, illustrated with pictures, however, it has very limited scientific content and its contribution to the literature is not clear.

Response: Thank you for your feedback.

My comment was only a polite statement for the authors, as the manuscript does not meet the expectations of a scientific publication, as there is no research question, there is no methodology (the Methods part is a simple descripive section) and the results are only the summary of the project implementation. 

The manuscript might fit to a case study collection, but not to a peer reviewed scientific journal.

Response: Thank you very much for your comment.

Reviewer 3 Report

Interesting case study. I would appreciate more detailed description of the food insecurity fears including literature citations. Second, I suggest to add a graph with numbers of delivered packages or families served over the project time - possibly corresponding with figure 1.

Reviewer 4 Report

The authors describe a program to address food insecurity during the first year and a half of the COVID-19 pandemic in South Florida and discuss the lessons learned from the intervention. The program appears to have potentially been effective in alleviating food insecurity in the population and the lessons learned are valuable. Nevertheless, I have several comments which I would urge the authors to consider. 

I would consider being a little more clear in the aims of the study. For instance, what intervention outcomes will be evaluated? This would help to frame the following text in the methods, results, and discussion sections, to help the text flow more logically. More importantly, it would help to be more clear about the generalization of the lessons learned. I understand that the intervention is focused on a specific community (Miami Dade County) and a specific crisis (COVID-19). I wonder though if the authors would be able to devote some text to discussing whether the lessons learned may generalize to other communities and exogenous events affecting food security. For instance, other communities may have had different resources such as access to transportation, grocery delivery, and programs to address poverty and food insecurity. Furthermore, I wonder whether some of the lessons learned may be applicable to other events such as economic recessions, which also negatively impact food insecurity. This information would make the contribution of the study much clearer.

I would also like a little more information on the sample. The descriptive table is currently indicated as supplemental text, but this should definitely be brought into the main text. Furthermore, is information available on how many people in the sample lost jobs or had their hours cut? Other relevant descriptive information would also be welcome (household size, whether respondents are parents and how many children are in the home, whether one got COVID, level of education, enrollment in other programs to supplement income or alleviate food insecurity).

It would also be very helpful to discuss how food insecurity was measured. The text states that "household members who reported food insecurity doubled from 3.5 to 8.0 percent", but the authors should indicate the specific question or definition of food insecurity that was presented to participants.

I was happy to see that food boxes were designed to be culturally appropriate, but readers may appreciate a little more detail on what went into the culturally appropriate food boxes.

Last, in the limitations section, the authors mention that the project "was able to decrease the level of food insecurity from its peak 39% to 23% when the project ended." I would encourage the authors not to use causal language here. It is plausible that the reduction in food insecurity may have been due to easing of stay at home orders and returning to work as the pandemic progressed.

Round 2

Reviewer 1 Report

The Authors have addressed my comments and/or corrected inaccuracies. In addition, the text has been expanded in several places in the paper to better clarify important points. Moving Table 1 from the Supplementary Material to the paper is also a good solution. However, if this is done, the Supplementary Material should be removed and not referenced in the paper (remove the subsection from lines 327-328). Also, the font in Table 1 should be smaller and its title should be placed above it, not below it.

Author Response

Reviewer’s comments: The Authors have addressed my comments and/or corrected inaccuracies. In addition, the text has been expanded in several places in the paper to better clarify important points.

Response: Thank you, your feedback has significantly improved the manuscript.

Reviewer’s comments: Moving Table 1 from the Supplementary Material to the paper is also a good solution. However, if this is done, the Supplementary Material should be removed and not referenced in the paper (remove the subsection from lines 327-328).

Response: Thank you. The text alluding to the supplementary material has been deleted.

Reviewer’s comments: Also, the font in Table 1 should be smaller and its title should be placed above it, not below it.

Response: Thank you for the feedback. The font in table 1 has been changed to Palatino Linotype, size 10. Line 258-259.

Reviewer 2 Report

Even though the manuscript has substantially improved, my basic concerns provided earlier remained.

Author Response

Reviewer 2: Comments and Authors’ Response

Reviewer’s comments: Even though the manuscript has substantially improved, my basic concerns provided earlier remained.

Earlier concern is as follows:

This is a nice description of a very interesting program, illustrated with pictures, however, it has very limited scientific content and its contribution to the literature is not clear.

Response: Thank you for your feedback.

My comment was only a polite statement for the authors, as the manuscript does not meet the expectations of a scientific publication, as there is no research question, there is no methodology (the Methods part is a simple descripive section) and the results are only the summary of the project implementation.

The manuscript might fit to a case study collection, but not to a peer reviewed scientific journal.

Response: Thank you for your feedback. I am not sure there is comment to address.

Reviewer 4 Report

The authors addressed my concerns with the manuscript. I have no additional comments.

Author Response

Reviewer’s comments: The authors addressed my concerns with the manuscript. I have no additional comments.

Response: Thank your for your guidance, we believe it has significantly improved the manuscript.